# Honey bee sting pain index by body location

Michael L. Smith

Department of Neurobiology and Behavior, Cornell University, USA

## ABSTRACT

The Schmidt Sting Pain Index rates the painfulness of 78 Hymenoptera species, using the honey bee as a reference point. However, the question of how sting painfulness varies depending on body location remains unanswered. This study rated the painfulness of honey bee stings over 25 body locations in one subject (the author). Pain was rated on a 1–10 scale, relative to an internal standard, the forearm. In the single subject, pain ratings were consistent over three repetitions. Sting location was a significant predictor of the pain rating in a linear model ($p < 0.0001$, $DF = 25, 94$, $F = 27.4$). The three least painful locations were the skull, middle toe tip, and upper arm (all scoring a 2.3). The three most painful locations were the nostril, upper lip, and penis shaft (9.0, 8.7, and 7.3, respectively). This study provides an index of how the painfulness of a honey bee sting varies depending on body location.

## INTRODUCTION

Popularly called the Schmidt Sting Pain Index (*Berenbaum, 2003*; *Schmidt sting pain index, 2013*), Justin Schmidt judged the painfulness of stings from 78 species of Hymenoptera (*Schmidt, Blum & Overal, 1984*; *Schmidt, 1986*; *Schmidt, 1990*). Schmidt's 4-point scale ranges from 0, a sting that cannot penetrate the skin, to 4, the most painful insect sting known (*Schmidt, 1990*). Only the bullet ant, *Paraponera clavata*, and the tarantula hawk, *Pepsis grossa*, were awarded a painfulness of 4 (*Schmidt, 1990*). A honey bee sting, *Apis mellifera*, which most people have experienced, was given a rating of 2 (*Schmidt, 1990*). It was used as an internal standard: a base point for investigators to rate the other stings. Knowing how sting painfulness varies among species is useful, but how sting location influences pain remained unanswered. *Schmidt (1986)* recognized this, commenting that: "pain levels from particular stings do, of course, vary and depend on such features as where the sting occurred (. . . )" (*Schmidt, 1986*). However, they lacked a model for understanding how pain varies depending on sting location.

If sting location is important in pain perception, how important is it? For example, are certain sting locations more painful than others? Which locations are the most painful and which are the least? To address this question of location-based pain perception, one requires a standard stimulus. The present study used the sting of the European honey bee (*Apis mellifera*) as its standard. A sting from a honey bee is familiar to many because of its world-wide distribution. The sting can be reliably provoked, and standardized, making it

Corresponding author
Michael L. Smith,
mls453@cornell.edu

an ideal experimental stimulus. Furthermore, its rating as the center point of the Schmidt pain scale suggests it may be a useful standard. The present study therefore used honey bee stings to determine whether sting location impacts painfulness, and how painfulness varies by location.

Pain is notoriously difficult to quantify. Many pain-rating scales have been developed to bridge the gap between a patient's perceived pain, and the medical practitioner who is trying to relieve the patient's pain. Different scales use numerical ratings, verbal ratings, questionnaires, visual depictions, or a combination of these measurements (see review by *Williamson & Hoggart, 2005*). In practice, each scale has its own advantages and disadvantages. Experimental testing of four commonly used pain scales found that all were valid, with low variability within a single scale (*Ferreira-Valente, Pais-Ribeiro & Jensen, 2011*). Previous research also found that numerical rating scales are the most responsive, relative to other pain scales (*Ferreira-Valente, Pais-Ribeiro & Jensen, 2011*). This study used a numerical rating scale, to simplify comparisons between stings at different body locations.

Using a similar pain stimulus (a honey bee sting), this study attempts to map the painfulness of stings according to where on the body the sting occurs. The purpose of this study is to rate the painfulness of honey bee stings over the human body, using a single subject (the author).

## MATERIALS AND METHODS

Cornell University's Human Research Protection Program does not have a policy regarding researcher self-experimentation, so this research was not subject to review from their offices. The methods do not conflict with the Helsinki Declaration of 1975, revised in 1983. The author was the only person stung, was aware of all associated risks therein, gave his consent, and is aware that these results will be made public.

Twenty-five sting locations were selected throughout the body (see Fig. 1). One location (forearm) was selected as an internal standard, with the a priori assumption that stings to the forearm would induce a median level of pain. The author self-administered five stings per day. The first sting and last sting were the internal standards (forearm). These stings were given a score of "5", and the three "test" stings were rated relative to the pain of the forearm stings. All stings occurred between 0900 h and 1000 h, to avoid time of day effects. At least 5 min of delay was given between stings, longer if pain from the previous sting persisted. The pain was rated by the author as precisely as possible on a scale of 1–10, relative to the internal standard (score of 5). Lower scores denote less pain; higher scores denote more pain. A numerical rating scale was used to simplify comparisons between sting locations. Previous research has found that numerical rating scales are the most responsive relative to other pain scales (*Ferreira-Valente, Pais-Ribeiro & Jensen, 2011*).

Sting locations were randomly ordered by the statistical program R (*R Core Team, 2012*). When applicable, the left and right side of the body were alternated. Some locations required the use of a mirror and an erect posture during stinging (e.g., buttocks). Stinging occurred before the author did any other honey bee work, to prevent unintentional stings

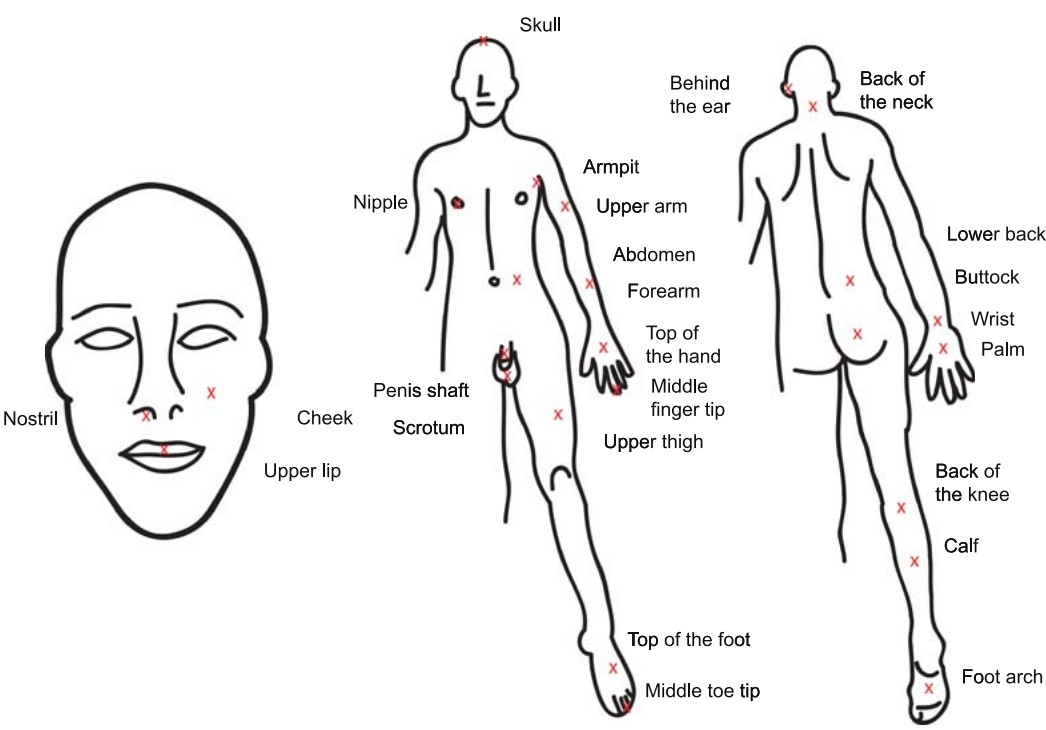

**Figure 1** **Sting Locations.** Drawing of the human form with Xs and labels at the sting locations.

during routine bee work from interfering with the experimental stings. The author had received approximately 5 stings per day for three months before the experiment, so no changes in his immune system were to be expected over the course of the experiment (*Light et al., 1975*).

Synthetic melittin was not used because this study focuses on natural honey bee stings. Although it is impossible to fully standardize natural honey bee stings, the following measures were used. Honey bees were collected from the hive entrance and only guards were selected, as identified by their stance (*Seeley, 1985*). Guarding is a specialized defensive task (*Moore, Breed & Moor, 1987*), so these bees are more likely to sting under natural settings (*Breed, Guzmán-Novoa & Hunt, 2004*). Guard bees were collected in a cage, and used immediately. Bees were taken from the cage haphazardly with forceps. To apply the sting, the bee was grabbed by the wings and pressed against the desired sting location. The bee was held against the sting location until the sting was first felt, and kept at the location for 5 s to ensure that the stinger would penetrate the skin. The bee was pulled away after 5 s, leaving the stinger in the skin. The stinger was left in the skin for 1 min, and then removed with forceps.

In total, three full stinging rounds were conducted at the Liddell Field Station of Cornell University in Ithaca, New York (42°27.6′N, 76°26.7′W). The author was stung over a total of 38 days, between 20 August 2012 and 26 September 2012. To keep the author as blind to the ratings as possible, notes were kept hidden from previous days. After two stinging rounds had been conducted (each stinging round covered all anatomical sting locations),

the scores were reviewed, to see if there was a large discrepancy between scorings per sting location. Only one location differed by 3 units (foot arch), and two locations by 2 units (upper thigh and behind the ear). Even though the consistency between the first two rounds was high, a third round of stinging was performed.

Statistical tests and linear models were analyzed using R, and the packages stats and lme4 (*R Core Team, 2012*). Descriptive statistics (mean and standard deviation) were summarized for each body location. A linear model was constructed after checking that the data satisfied all linear model assumptions: normality of residuals, constant variance of errors, and no serial correlation. The linear models were used to determine if there were any significant differences in pain rating (response variable) according to the body location, the side of the body, the stinging round, or the stinging date (predictor variables). Models were compared using AIC values, with lower AIC values indicating a better fit.

All the pain ratings collected in this experiment came from one person (the author), to minimize the number of people stung. Statistical testing was used to describe the results, and determine if body location would predict the pain rating. However, because only one person was stung, these data are repeated measures of a single subject ($n = 1$). The data should therefore be taken to represent only this person, and not be generalized for the public.

## RESULTS

All the stings induced pain in the author. The pain rating for each location was averaged over the three rounds, and ordered from lowest to highest (see Table 1). The three least painful locations were the skull, middle toe tip, and upper arm (all scoring a 2.3). The three most painful locations were the nostril, upper lip, and penis shaft (9.0, 8.7, and 7.3, respectively).

Sting location was a significant predictor of the pain rating in a linear model ($p < 0.0001$, $DF = 25, 94$, $F = 27.4$). Whether the sting was on the left or right side of the body was not significant ($p = 0.58$, $DF = 1, 106$, $F = 0.30$). The stinging round (1st, 2nd, 3rd) was not significant ($p = 0.90$, $DF = 2, 117$, $F = 0.12$). Stinging date was not significant ($p = 0.92$, $DF = 23, 96$, $F = 0.59$). Comparing the sting location model, which only takes into account sting location, with a null model, the RSS was significantly lower in the sting location model ($RSS = 34.00$ vs $281.97$, $p < 0.0001$), and the AIC values were lower in the sting location model (243 vs 447). If stinging date and stinging round were added to the sting location model, the AIC increased from 243 to 252, so they were not included in the model.

## DISCUSSION

It was found that the location of the sting had a significant impact on the level of perceived pain in the single subject tested. Sting location was a significant predictor of the pain felt (see results). This result is expected; a honey bee sting is expected to induce more pain depending on where it occurred.

The variability of pain perceived at a given site was low, often only a unit or two between the three rounds (see Table S1). This suggests that pain receptors are uniformly sensitive in

**Table 1** Average pain ratings.

| Body location | | Pain rating | |
| --- | --- | --- | --- |
| **Medical terminology** | **Layperson terminology** | **Average rating** | **Standard deviation** |
| Anterior vertex | Skull | 2.3 | 0.6 |
| Third distal phalanges (foot) | Middle toe tip | 2.3 | 0.6 |
| Proximal humerus, dorsal aspect | Upper arm | 2.3 | 0.6 |
| Buttocks | Buttock | 3.7 | 0.6 |
| Dorsal aspect of leg | Calf | 3.7 | 0.6 |
| Posterior trunk, lumbar region | Lower back | 4.0 | 1.7 |
| Anterior aspect of proximal thigh | Upper thigh | 4.7 | 1.2 |
| Anatomic wrist, ventral aspect | Wrist | 4.7 | 0.6 |
| Foot, plantar surface | Foot arch | 5.0 | 1.7 |
| Distal arm, dorsal aspect | Forearm | 5.0 | n.a. |
| Popliteal fossa | Back of the knee | 5.0 | 1.0 |
| Posterior neck, cervical region | Back of the neck | 5.3 | 1.2 |
| Postauricular | Behind the ear | 5.3 | 1.2 |
| Hand, dorsal aspect | Top of the hand | 5.3 | 1.2 |
| Foot, dorsal aspect | Top of the foot | 6.0 | 1.0 |
| Abdomen | Abdomen | 6.7 | 0.6 |
| Third distal phalanges | Middle finger tip | 6.7 | 0.6 |
| Nipple | Nipple | 6.7 | 0.6 |
| Axilla | Armpit | 7.0 | 0.0 |
| Buccal aspect of face | Cheek | 7.0 | 0.0 |
| Hand, anterior aspect | Palm | 7.0 | 0.0 |
| Scrotum | Scrotum | 7.0 | 0.0 |
| Body of penis, dorsal aspect | Penis shaft | 7.3 | 0.6 |
| Tubercle of superior lip | Upper lip | 8.7 | 0.6 |
| Anterior nares | Nostril | 9.0 | 0.0 |

a given body area, contrary to the common saying: "hitting a nerve". Again, this result only holds for the single subject tested.

The controls in this honey bee sting pain index closely mirror results from pressure pain studies. For example, there was no difference in pain perception between the left and right side of the body, as reported previously for pressure pain response (*Fischer, 1986*; *Fischer, 1987*). It is unlikely that the author habituated to stings over the experiment, because both the date of stinging, and the stinging round, were not significant predictors of the pain perceived. These control results match previous work in pressure pain response, but do not explain why certain locations were the most painful to honey bee stings.

The three most painful sting locations were the nostril, the upper lip, and the penis shaft (average pain scores of 9, 8.7, and 7.3, respectively) (see Table 1). The three least painful locations were the skull, middle toe tip, and upper arm, all scoring a 2.3. Why were certain locations more or less painful? For the most painful locations, sting depth may be important, because the skin is thinnest on the genitals, followed by the face (*Ya-Xian, Suetake & Tagami, 1999*). The nose and lips are orifices, so they may also have lower

pain thresholds for protection. Stings to the nostril were especially violent, immediately inducing sneezing, tears and a copious flow of mucus. The sting did autotomize in the nostril (self-severed when the bee was pulled away). The copious mucus flow, however, may help prevent subsequent stings to the area during a natural attack.

Skin thickness, however, does not fully explain how painful a location will be. For example, the palm of the hand has twice as many skin layers as the dorsum (*Ya-Xian, Suetake & Tagami, 1999*), but the palm received a pain score of 7.0, and the dorsum, a 5.3. Furthermore, the least painful locations did not have the largest number of skin layers- the skull and upper arm have approximately one fourth the number of skin layers as the palm (*Ya-Xian, Suetake & Tagami, 1999*). Clearly, skin thickness is not the only factor for predicting painfulness. Perhaps receptor thresholds are lower depending on the 'importance' of certain locations, or the CNS reaction is amplified depending on the location of the sting.

The somatosensory homunculus, a drawing of the human form scaled according to cortical area devoted to that body region, could explain which areas are more sensitive, although it focuses on mechanosensory neurons. The tongue and thumb, followed by the lips and digits, are the most sensitive areas, as measured by neural activity (*Hämäläinen, Hari & Ilmoniemi, 1993*; *Nakamura et al., 1998*). The present study did not sting the tongue or the thumb, but the upper lip *was* one of the most painful locations. However, the neural activity range for the upper and lower lip overlaps with that of the middle finger, suggesting that all three locations would have similar sensitivity. In this pain index, the middle finger and the upper lip were not similar, scoring a 6.7 and 8.7, respectively. The differences in sensitivity suggest that even if neurons map to specific body locations, they may not be the same for different sensory information. It is plausible that a 'pain' homunculus would look different from a somatosensory homunculus.

This study is limited by its low sample size: one person, the author. It is possible that if other people were tested, they would not rank the painfulness of the stings in the same way, or perceive pain similarly by location. Although these findings cannot be generalized, they are still interesting. Some locations only apply to male anatomy (i.e., scrotum and penis), and males are known to have differing pain thresholds compared to females (*Berkley, 1997*). This index is only meant as a first approximation of how sting pain varies by location. In support of this pain index, the pain ratings per location were similar between the three rounds (see methods and Supplemental Information). This suggests that the index is accurate for rating sting painfulness by body location in the single subject. Other factors, such as sting duration (*Visscher, Vetter & Camazine, 1996*), and sting depth, would also influence pain perception (*Topazian, 1957*).

This experiment serves as an orthogonal extension of the Schmidt Sting Pain Index, and a rough map for painfulness based on body location.

## ACKNOWLEDGEMENTS

Painful appreciation to Justin Schmidt, Kevin Loope, Paul Shamble, Jason Barry (Cornell Statistical Consulting Unit), Scott Camazine, and Tom Seeley for helping design the

experiment and edit the manuscript. Thanks to the editor and anonymous reviewers for helpful critique that greatly improved the paper. Special thanks to Peter Dunbar for engaging medical discussion.

### Funding

This material is based on work supported by a United States National Science Foundation (NSF) Graduate Research Fellowship. The funders had no role in study design, data collection or analysis, decision to publish, or preparation of the manuscript.

### Grant Disclosures

The following grant information was disclosed by the author:
United States National Science Foundation (NSF) Graduate Research Fellowship.

### Competing Interests

The author declares there are no competing interests.

### Author Contributions

- Michael L. Smith conceived and designed the experiments, performed the experiments, analyzed the data, contributed reagents/materials/analysis tools, wrote the paper, prepared figures and/or tables, reviewed drafts of the paper, and was the experimental subject.

### Ethics

The following information was supplied relating to ethical approvals (i.e., approving body and any reference numbers):

Cornell University's Human Research Protection Program does not have a policy regarding researcher self-experimentation, so this research was not subject to review from their offices. The methods do not conflict with the Helsinki Declaration of 1975, revised in 1983. The author was the only person stung, was aware of all associated risks therein, gave his consent, and is aware that these results will be made public.

### Supplemental Information

Supplemental information for this article can be found online at http://dx.doi.org/10.7717/peerj.338.

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
