# Peer review of "Honey bee sting pain index by body location"

_PeerJ, doi:10.7717/peerj.338_

## Round 0.1 · original submission · Minor Revisions

Dr. Smith thank you for you submission. After considerable deliberation, I believe your manuscript will be suitable for publication with revision. I think the comments of the reviewers on individual points are helpful and merit consideration. It is perhaps harder to address the more general issues raised in the reviews. Both reviewers had questions regarding the suitability of the statistical analysis, and I did also. Ultimately, I formed an opinion on this issue, but before communicating with you, I discussed this with Dr. Peter Binfield (Pete is one of the co-founders and publishers of PeerJ ) to make certain my interpretations were consistent with PeerJ policies.

The following is a copy of my email and Pete's responses, so you can see our thinking:

"I worked on this at length yesterday, and needed to ask your advice relative to PeerJ policies, etc.

The reviewers both recommended acceptance but both with serious reservations about whether or not the study is statistically valid. The core issue is that this study is essentially observational and the "replicates" of measures of pain index could be interpreted as repeated measures on a single subject (because only one subject - the author - provided data). A further complication is that the pain index used is a variation of a previous index on insect stings which had no relationship to other (published) indices on measuring pain. In my (limited) review of the medical literature on pain measurement, it is clear that no objective or physiological measures of pain exist, but there is a considerable literature on approaches to measurement and to mechanisms associated with different types of pain. As a further consideration, few studies on pain from insect stings exist, and it would not surprise me that this study, if published, would attract media attention (for good or bad).

Relative to PeerJ, I think the issue is how we handle observational or case-study papers. In my work with in forensic science and (more limited) experience with medical literature, I've been of the opinion that standards for case study publication are (broadly) too liberal and too vague (but I'm mostly an experimentalist, so I may be biased here). I think observations of novel phenomena have an important place in the scientific literature, and sets of observations can provide information not available through other sources. So these have emerged as my personal standards. I think this study meets that standard.

Regarding statistics, this pain study could have been conducted with multiple subjects, with each subject representing an experimental unit, and there would be little question about statistical validity. However, there is ample president for not taking such an approach with certain types of treatments, especially when the experimentation is on humans. I think the most convincing example I'm aware of is the work by Marshall and Warren establishing that Helicobacter (formerly Campliobacer) pylori is the cause of gastric ulcers. In particular, the Marshall et al. paper of 1985 (attached) in which Marshall ingested H. pylori and developed gastristis within days (!) is often cited as providing compelling evidence. In this work the n=1, Marshall was the only subject, and a good thing too, I'd say. Anyway, as Marshall and Warren received the Nobel for their work, I'd say the scientific community has endorsed single subject studies in certain circumstances.

Accepting the arguments above (for publication) I still find the validation of the pain index problematic, and I would like the author to link his index to other approaches. I'd also like to see greater review of medical pain literature in the introduction or discussion section. It seems one thing this paper could achieve is to make the entomological and ecological communities more aware of that literature, through a short literature review, however, I don't know if such a recommendation is in keeping with PeerJ policies.

So my questions for you are: one, do we have a PeerJ policy on observational/case studies or does decision-making reside with the subject editor? Second, is it appropriate for me to recommend adding a more extended literature review in this instance (beyond just the literature on insect stings and pain)?"

Pete replied:

We don’t allow case reports in the journal, but this is mainly to avoid the pure medicine case report of “random patient presented with random problem and it was/wasn’t solved in this one case by doing X” – in those cases the study wasn’t pre-planned; there was no attempt (or opportunity) for replication; the results can’t be called significant etc. In this case though, the study was at least pre-planned, and each sting was administered several times etc, so I think it is OK from the ‘case report’ point of view.

As you note though, it would have been better to have done the study on more than one person of course. The bottom line, is that I think it is OK (according to our policies) and so you should then evaluate it on the content / research. It is fine to ask for more substantial literature description (we don’t allow pure literature reviews, but inclusion of a literature review inside an otherwise acceptable article is fine).

So, the bottom line is that I think a study on pain from insect stings, falls into one of those categories in which data from a single subject appropriately collected (as you did) is a legitimate scientific contribution. As I mention above, I think it will strengthen your paper (and possibly deflect potential criticism) to directly address this issue in the Materials and Methods to justify the approach you used.

Regarding the pain scale you used, without research on human ability to reliably discriminate pain, I'm not sure how one justifies any scale or another. Often, in questions of quantifying subjective data, as is the case with many types of scaling, the issue is ignored. In my brief examination of the pain literature, I cannot find a good argument for one scale over another, and perhaps your perspectives on this issue would be valuable. My suggestion of adding a couple pages of "literature review" on pain and its measurement is to help justify your approach and, perhaps, better educate entomologists and ecologists. This suggestion is not a requirement for publication, but I do think it would strengthen your paper and it's usefulness. As an example, I wrote a computer program to calculate degree-days back in the 80's but I included a (long-winded) discussion of the theory behind the used of degree days to measure insect development. As it's turned out that has become one of the most cited papers in my career, so my thinking is that your manuscript may offer the same short of opportunity for you.

Before closing, I do apologize for the long delay getting back to you. Part of this was a combination of screwups and ill health on my part, and part was making certain that your manuscript received a thorough and fair review. Please don't hesitate to contact me if you have any questions regarding your revision -- I am looking forward to seeing your paper in PeerJ.

Reviewer 1 ·

Basic reporting

Article is generally well-written and interesting. Author should present variability in table 1 (standard error/ standard deviation).

Experimental design

Author recognizes that a single subject was used and analysis is approached rationally. I wonder about the standardization of 5 on a 10-point scale versus standardizing the most painful as the upper limit. It doesn't change the results but if a nostril is a "9" I wonder what a "10" from a honeybee would be. Secondly, this approach creates a "new" index different than the existing Schmidt Sting Pain which is a 4 point scale. The author should consider adding a column to table 1 converting his numbers to those of the existing index. Perhaps this will encourage someone to try a Pepsis sting on the nostril?

Second the author should describe in more detail how sting location was analyzed as a predictor of pain using a linear model. Location is relatively arbitrary not "predictive" (i.e. you know pain varies by location but a sting to the upper neck may or may not differ from the lower neck). Left versus right of the body could be as simple as a t-test.

Validity of the findings

Experiment well controlled and results appear valid. Do check the assumptions of analysis by a general linear model and add details (or revise). The data stand on their own and I am not sure a statistical analysis that differences are present adds very much,.

Additional comments

Interesting study overall. This is sure to become a classically cited study in general entomology classes.

Annotated reviews are not available for download in order to protect the identity of reviewers who chose to remain anonymous.

Reviewer 2 ·

Basic reporting

The manuscript provides information on differences in the Schmidt Sting Pain Index depending on the location of the sting on the human body. The information fits into the broader field of knowledge. Relevant literature is appropriately referenced, but other literature may be lacking (see below). The paper is fairly well written and needed changes to syntax, style, and grammar are minimal.

Experimental design

The experimental design, and especially the statistics, needs to be explained in MUCH more detail. I’m assuming the author stung himself, but that is not stated in the manuscript. More important, there are numerous potential issues with replication, experimental unit, and repeated measures that need to be carefully articulated to the reader because, as written, the design does not seem to be statistically valid. There is one experimental unit (the author), numerous locations on the body serving as “factors”, and three rounds of applying the stings to the areas/factors. The three rounds then would be repeated measures on the same experimental unit. They are not truly independent replicates. I do not have firsthand experience with research on very small sample sizes. Consequently, the methods may be appropriate, but the onus is on the author to thoroughly describe the methods. Regardless of whether the index measures of pain are true replicates, the author does not discuss whether the discrete pain ratings meet assumptions of normality. Ratings often are non-parametric and therefore the author’s analysis may be invalid.

Validity of the findings

See above for more information on statistical validity.

How do the sting locations vary with respect to specific nerve locations? Could the differences in index be the result of more often hitting a nerve in one location as opposed to missing it in another location? The paragraph on somatosensory homunculus (lines 148-159) is filled with overly technical jargon that needs to be defined for a broader audience.

Specific Comments:
Line 24. Replace “rated” with “rates” and “hymenoptera” with “Hymenoptera”.
Line 25. Replace “varied” with “varies”.
Line 26. Replace “remained” to “remains”.
Line 35. Italicize “Apis mellifera”.
Line 41. Common names of insects should be lowercase unless the common name includes a formal noun.
Line 43. Replace hyphen after “standard” with a colon.
Line 46. Add the year to “Schmidt et al.”
Line 64. Replace “25” with “Twenty-five”.
Line 68. Replace “was” with “were”.
Line 69. The hyphen is not the appropriate punctuation point.
Line 73. “All sting locations were performed” is very awkward phrasing. Rewrite.
Line 76. Replace “prior to” with “before”.
Line 80. Replace “While” with “Although”.
Lines 119-121. This sentence needs supporting citations.
Line 129. Replace “but it does not” with “but do not”.
Line 144. The phrase “4-times fewer” is not mathematically possible. Any number multiplied by four cannot be less than the number being multiplied by 4. Rewrite to emphasis that those locations have one-fourth the skin layers than the palm.
Line 155-156. Be careful using the term “equivalent”. In most cases, the relevant term is “not significantly different” or “similar”. Equivalent implies exactly the same, which cannot be tested statistically.
Line 176. The first sentence is confusing and unnecessary. Delete.
Table 1. Why isn’t variability expressed in this table? Presentation of an average rating is of limited value without some indication of variability.

---

## Round 0.2 · accepted · Accept

Dr. Smith, I'm happy to say your manuscript is suitable for publication. I think you did an outstanding job with your revision, both in addressing reviewer comments and my own concerns. I was particularly impressed with your revised discussion on pain scales and how you referenced the pain literature. Most importantly, I found your discussion of the significance of your study to fit exactly the right balance. I'm very pleased to see how well this version works, and I'm delighted that you chose PeerJ as an outlet for your work.